# Lung-Heart Outcomes and Mortality through the 2020 COVID-19 Pandemic in a Prospective Cohort of Breast Cancer Radiotherapy Patients

**DOI:** 10.3390/cancers14246241

**Published:** 2022-12-18

**Authors:** Vincent Vinh-Hung, Olena Gorobets, Nele Adriaenssens, Hilde Van Parijs, Guy Storme, Dirk Verellen, Nam P. Nguyen, Nicolas Magne, Mark De Ridder

**Affiliations:** 1Department of Radiation Oncology, Centre Hospitalier de la Polynesie Française, Pirae 98716, French Polynesia; 2Oncology Center, Universitair Ziekenhuis (UZ) Brussel, 1090 Brussels, Belgium; 3Department of Radiation Oncology, Institut Bergonié, 33076 Bordeaux, France; 4Department of Maxillofacial Surgery, Centre Hospitalier Universitaire de Martinique, 97213 Le Lamentin, France; 5Iridium Network and Faculty of Medicine and Health Sciences, Antwerp University, 2000 Antwerp, Belgium; 6Department of Radiation Oncology, Howard University, 2041 Georgia Ave NW, Washington, DC 20060, USA

**Keywords:** COVID-19, longitudinal analysis, cohort monitoring, outlier’s test, hybrid prospective-retrospective study, adjuvant radiotherapy, pulmonary function test, aging

## Abstract

**Simple Summary:**

A cohort of 123 women with stage I-II breast cancer enrolled in a prospective clinical trial of adjuvant radiotherapy who were followed for over 10 years. Year 2020 was associated with excess mortality. The number of deaths (*n* = 5) in that single year represented a third of all deaths (*n* = 15) in the full decade before. There was in that year a significant increase in lung-heart toxicity and a significant decline in lung-heart function as measured by left ventricular ejection fraction, forced vital capacity and carbon-monoxide diffusing capacity.

**Abstract:**

We investigated lung-heart toxicity and mortality in 123 women with stage I-II breast cancer enrolled in 2007–2011 in a prospective trial of adjuvant radiotherapy (TomoBreast). We were concerned whether the COVID-19 pandemic affected the outcomes. All patients were analyzed as a single cohort. Lung-heart status was reverse-scored as freedom from adverse-events (fAE) on a 1–5 scale. Left ventricular ejection fraction (LVEF) and pulmonary function tests were untransformed. Statistical analyses applied least-square regression to calendar-year aggregated data. The significance of outliers was determined using the Dixon and the Grubbs corrected tests. At 12.0 years median follow-up, 103 patients remained alive; 10-years overall survival was 87.8%. In 2007–2019, 15 patients died, of whom 11 were cancer-related deaths. In 2020, five patients died, none of whom from cancer. fAE and lung-heart function declined gradually over a decade through 2019, but deteriorated markedly in 2020: fAE dipped significantly from 4.6–4.6 to 4.3–4.2; LVEF dipped to 58.4% versus the expected 60.3% (PDixon = 0.021, PGrubbs = 0.054); forced vital capacity dipped to 2.4 L vs. 2.6 L (PDixon = 0.043, PGrubbs = 0.181); carbon-monoxide diffusing capacity dipped to 12.6 mL/min/mmHg vs. 15.2 (PDixon = 0.008, PGrubbs = 0.006). In conclusion, excess non-cancer mortality was observed in 2020. Deaths in that year totaled one-third of the deaths in the previous decade, and revealed observable lung-heart deterioration.

## 1. Introduction

The COVID-19 pandemic waves are unprecedented devastating disease outbreaks. We all have vivid memories of emergency departments and hospital wards overwhelmed with critical cases. Early on, caregivers were aware that cancer patients represented a particularly vulnerable population [1,2]. There is a need to learn the most from recent experiences [3,4]. However, longitudinal cohort studies quantifying the health status of cancer patients throughout the pandemic have been scarce [5,6], or were of short duration and were restricted to infected cases [7,8,9,10,11]. While the pandemic peaks appear to have dwindled, the prospect of future resurgences of COVID variants or the apparition of other virulent pathogens is a permanent societal concern. For cancer patients, it might be critical to identify whether their outcomes can be affected in pandemic circumstances. To that effect, the observation of patients who were previously enrolled in clinical trials, in whom assessments were prospectively collected, could provide valuable insight on the changes in adverse health effects.

Such is the case with the TomoBreast clinical trial. Early meta-analyses of breast cancer treatment data demonstrated a survival advantage by using adjuvant radiotherapy for breast cancer as part of a conservative strategy [12]. Based on the data, the researchers considered that breast cancer mortality reduction was conditional on cardio-pulmonary toxicity reduction. They hypothesized that advanced image-guided radiation treatment, as compared with conventional radiotherapy, could provide a clinically meaningful reduction of lung-heart toxicity in the post-surgery treatment of early and intermediate stage breast cancer, without adversely affecting tumor control and survival [13,14]. As core components of the trial, lung and heart function were purposefully assessed. Now in its 14th year, TomoBreast has gathered the largest known open database of longitudinal lung and heart function in breast cancer [15], making it uniquely relevant to investigate the impact of the pandemic.

Follow-up of the trial was continuous and unabated except for a disruption in 2020 due to the COVID-19 pandemic. Recent studies have observed disruptions in patient care during the lockdown period of the pandemic, including cancer-related care [16]. Such lockdown-related disruptions have been correlated with increased morbidity in various patient populations [17,18], suggesting that adverse outcomes may be indirectly related to COVID-19, especially in populations of patients with pre-existing vulnerabilities [19].

The present study aims to investigate up to and through the COVID-19 pandemic and its impact on mortality and lung-heart injuries in a breast adjuvant radiotherapy patient setting as part of the TomoBreast clinical trial.

## 2. Materials and Methods

### 2.1. Description of the Cohort

The study population includes participants in the TomoBreast clinical trial which randomized post-surgery stage I-II breast cancer patients into two groups: adjuvant image-guided radiotherapy vs. adjuvant conventional radiotherapy. The trial’s design, summary, flowchart and cardiopulmonary quality of life related results have been detailed elsewhere [14]. The trial started in June 2007 and ended in July 2011. A total of 123 women consented to participate (Table 1). The follow-up cutoff date was 16 July 2021. The collected data included patients’ self-reported outcome, clinical assessment, echocardiographic measurements, and pulmonary function tests, as detailed in the next sections. All patients were analyzed in the present study as a single cohort, irrespective of the randomization assignment.

### 2.2. Clinical Data Collection

Patients’ information was abstracted from individual patient’s electronic medical files and from the Belgian federal eHealth platform [20,21,22], at each date of medical contact related to breast cancer or any other condition. If available, clinical assessments done less than 365 days before randomization were also abstracted. The total number of medical records was 3013, averaging 24.5 per patient. The abstracted data included Karnofsky performance status (KPS), weight, and echocardiographic and pulmonary function test results.

### 2.3. Toxicity Grades

Heart and lung toxicities were recorded as the maximal score in any of the Radiation Therapy Oncology Group (RTOG) late morbidity or any of the Subjective Objective Management Analytic/Late Effects on Normal Tissues (SOMA/LENT) cardiac and pulmonary items. The toxicity grades were coded on an ordinal scale, from 0, i.e., G0, no toxicity, to 4, i.e., G4, worst grade.

### 2.4. Freedom from Adverse Event (fAE) Scores

The patients’ toxicity grades were converted into continuous “freedom from adverse event” (fAE) scores ranging from 1 (worst) to 5 (best) by applying fAE = (5—toxicity grade). Though not strictly necessary, the conversion allows the semantic interpretation of a positive slope (i.e., increasing fAE) as an indicator of health improvement, and a negative slope (i.e., decreasing fAE) as an indicator of deterioration.

### 2.5. Echocardiography Assessment

A total of 848 echocardiography reports were retrieved, averaging 6.9 per patient at different time points. Left ventricular ejection fraction (LVEF) was measured using Simpson’s biplane method. LVEF was reported in 830 exams and was omitted in 18 exams, 3 of which were estimated from the Teichholz method, 3 from Mitral Deceleration Time, and 1 from Tricuspid annular plane systolic excursion (TAPSE), leaving 11 echocardiographic records with truly missing LVEF. The LVEFs were originally reported as continuous exact numeric values in 615 reports, and as interval values of “50–55” in 1 report, “50–60” in 44 reports, and “>60” in 170 reports. The interval values were converted to a mid-interval value of 52.5 for “50–55”, to 55 ± 2 for “50–60”, and to 65 ± 2 for “>60”, taking into account other echocardiographic measurements specified in the reports. In total, 837 LVEF measurements were available.

### 2.6. Lung Function Assessment

Pulmonary function tests were retrieved from 796 reports, averaging 6.5 per patient at different time points. The measurements abstracted were forced vital capacity (FVC, *n* = 790 records), forced expiratory volume in the first second (FEV1, *n* = 796), peak expiratory flow (PEF, *n* = 785), vital capacity (VC, *n* = 694), total lung capacity (TLC, *n* = 693), residual volume (RV, *n* = 686 records), functional residual capacity (FRC, *n* = 679), airway resistance (Raw, *n* = 681), specific airway resistance (sRaw, *n* = 680), diffusing capacity for carbon monoxide (DLCO, *n* = 695), and alveolar gas volume (VA, *n* = 688). The direct measurements were used, without conversion to a percentage of predicted reference populations.

### 2.7. Interpolation and Handling of Missing Data

All clinical toxicity and echocardiography and pulmonary function test data were merged into a single data frame ordered by patient and by contact date, resulting in a full data frame with *n* = 4647 rows, with each row representing a unique patient time-record containing a toxicity score and a test measurement. Empty data cells were interpolated between non-missing time points. To illustrate with a fictional example, consider a patient who has, at day D20, an LVEF measurement but no adverse event (AE) score, at D30, an AE score but no LVEF measurement, and at D40, an LVEF but no AE score. The LVEF at D30 is interpolated from the D20 and D40 LVEFs; the AEs at D20 and D40 cannot be interpolated and remain missing.

### 2.8. Longitudinal Analysis of Continuous Measurements

The study’s general approach is that of a cohort monitoring over the calendar years with the intent of detecting adverse changes [23]. For the lung-heart outcomes, we already knew that the function steadily deteriorated over time [24]. Consequently, in order to detect whether an additional change occurred in the particular year 2020, we could not compare with a fixed time-point in the past but had to take into account what would have normally been expected if the pandemic did not occur. We applied ordinary least squares regression of toxicity grades (converted to fAE) and heart-pulmonary measurements over the years, excluding 2020. Thereafter we tested whether the actual measurements and toxicities observed in 2020 significantly departed from the regression as outliers. We used Dixon’s application of the Chi-squared test assuming the population variance is the same as that of the sample [25], and we used Grubbs’ corrected test [26].

### 2.9. Survival Analysis

The time variable required for survival analysis is the interval between an observation date and a given time origin, i.e., the follow-up year is embedded in the survival model. Unless all patients in a cohort are accrued on the same date, in which case the follow-up period equates the time elapsed, conventional survival analysis cannot detect the effect of a particular follow-up year. In the present study, we simply tabulated the deaths observed over the calendar years, without significance testing. Mortality in the study refers to the crude number of deaths.

For discussion purposes—not as study objectives—the overall survival from the date of randomization to death of any cause, and the breast cancer specific survival, from the date of randomization to the date of death from breast cancer, were computed using the Kaplan-Meier product-limit method [27].

### 2.10. Statistical Implementation

All statistical computations were implemented using R version 4.1.2 [28]. Packages used were outliers, survival, survminer and ggplot2. The complete implementation script is made available together with the shared data at https://zenodo.org/deposit/5919956 (accessed on 30 January 2022) with reserved doi:10.5281/zenodo.5919956.

## 3. Results

At the cutoff date, 103 out of the initial 123 patients enrolled in the study remained alive. All these 103 patients had a follow-up from randomization exceeding 10 years: the range extended from 10.03 to 13.99 years, with a median follow-up period of 12.04 years. Among the 20 patients who died, the median time to death was 8.3 years (range 3.6, 12.4). The calendar years of death were 2011 (*n* = 1 death), 2012 (*n* = 1), 2013 (*n* = 2), 2014 (*n* = 1), 2015 (*n* = 3), 2017 (*n* = 2), 2018 (*n* = 1), 2019 (*n* = 4), 2020 (*n* = 4), and 2021 (*n* = 1, pulmonary hypertension and right ventricular failure in October 2020). None of the five patients who died in 2020–2021 had any evidence of cancer (Table 2). Altogether, the predominant pathology at the times preceding death were advanced metastatic disease in 11 patients (nine with breast cancer, one of whom also had overt heart failure, one with new primary lung cancer, and one with new primary ovarian cancer), and various diseases, without evidence of cancer in the other nine patients (four with digestive pathology or liver failure, two with pulmonary failure, one with renal abscess, one with toxic thyroid adenoma, and one with an unknown cause that was ascertained by the patient’s general practitioner as non-cancer). With the exception of the one patient who died in 2013, cardiac pathology did not preponderantly contribute to death in any of these cases.

The cardiac and pulmonary toxicities observed throughout the 14 years of follow-up were generally mild and infrequent (Table 3). By the average of the total of years, there was no sign of any heart toxicity in 90.2% of the patient observations, and no sign of any lung toxicity in 79.5% (Table 3, bottom row). Grade 3 and 4 heart toxicities were rare (1.2% and 0.4%, respectively), as were Grade 3 and 4 lung toxicities (2.1% and 0.3%, respectively) (Table 3, bottom row). However, when compared by calendar year, gradual changes in the incidence of toxicities were observable.

Figure 1 graphically displays the information of Table 3 to facilitate reading. For heart toxicity, the global trends are a gradual but steady decline in the percentage of patients without heart toxicity over the years (Figure 1, Heart panel G0), no change in the Grade 1 percentage, and a small increase in the percentage of patients with Grade 2 to 4 heart toxicities (Figure 1, Heart panels G2, G3, G4). For lung toxicity, the global trends are that there was no significant change in the percentage of patients without lung toxicity (Figure 1, Lung panel G0), there was a decline in the percentage of patients with Grade 1 lung toxicity (Figure 1, Lung panel G1), and there was a small increase in the percentage of patients with Grade 2 to 4 lung toxicities (Figure 1, Lung panels G2, G3, G4).

The incidence of cardiac and pulmonary toxicities in 2020 is highlighted in Table 3 and Figure 1. Due to the COVID-19 lockdowns and contact restrictions, the number of analyzable heart and lung observations declined dramatically, from more than 1 per patient in the preceding years, to less than 1 in 2020 (Table 3, columns “*n* data”). The percentages of patients without heart or lung toxicity dipped markedly to 73.3% and 65.8%, respectively (Table 3, column G0). The Grade 0 dips seen in 2020 corresponded to unequally distributed peaks, mostly among the Grade 2–4 toxicities (Figure 1). From Table 3 in 2020, the pooled percentage of Grade 2–4 heart toxicities was 22.2% (=12.2 + 2.2 + 7.8) and lung toxicities was 28.7% (=19.2 + 6.8 + 2.7), which reflects the poor conditions of the few patients who did receive medical contact.

Reverse modeling the toxicity grade as a continuous “freedom from adverse event” (fAE) score that ranges from 1 (worst, severe toxicity) to 5 (best, fully free from any toxicity indicators), and pooling the fAE as a single variable (shown in Figure 2) indicates a small yet evident degradation of the yearly average fAE scores over time, for heart as well for lung toxicity. Year 2020 was associated with a further dip in heart and lung freedom from toxicity (Figure 2, plain circles). For the year 2020, the Grubbs and the Dixon-adapted chi-square tests indicated that departure from the regression line was significant for heart (PGrubbs = 0.046; PDixon = 0.019) and for lung (PGrubbs = 0.056; PDixon = 0.021) toxicities.

Similar to the patterns of toxicities, the heart and lung function tests deteriorated over time. The heart LVEF, the lung FVC, and the lung DLCO decreased, while the lung RV increased (Table 4, Figure 3). The LVEF in 2020 was 58.4%, presenting a significant dip compared with the expected LVEF of 60.3% (PGrubbs = 0.054; PDixon = 0.021). The FVC also presented a dip in 2020, at 2.4 L versus the expected value of 2.6 L (PGrubbs = 0.181; PDixon = 0.043). The DLCO presented a significant and profound dip at 12.6 mL/min/mmHg in 2020, versus the expected DLCO of 15.2 mL/min/mmHg (PGrubbs = 0.006; PDixon = 0.008). Other lung function tests, including the FEV1, PEF, VC, and FRC, also showed significant deterioration in 2020. However, lung function deterioration was not detected by the TLC, Raw, sRaw, and VA, the corresponding results are not shown but can be retrieved from the shared data as per the Data Availability Statement.

## 4. Discussion

Against a decade-long backdrop of gradual functional decline, year 2020 was associated with a peak deterioration in almost all cardiopulmonary indicators (Table 3 and Table 4). Year 2020 was also marked by an abrupt change in the patterns of dying. Up until 2019, deaths were sporadic and were cancer-related (Table 2). Immediately thereafter, coinciding with the pandemic, five deaths occurred in a row. None were cancer related. These deaths occurred within an interval of barely 1.5 years but represented 25% of all deaths among the trial’s patients. Intriguingly, no SARS-CoV-2 infections were reported for any of these deaths.

The pattern of deaths appears to mirror the severe excess mortality that affected the general population early in the COVID-19 epidemic [29]. A study of the Belgian Cancer Registry for the first half of 2020 estimated a 33% rise in mortality in April [30]. Crude mortality rates derived from Statbel, the Belgian statistical office, showed among women, a rate of 1096 deaths per 100,000 in 2020; i.e., an excess of 101 to 161 deaths per 100,000 as compared with the 2010–2019 rate of 935 to 995 deaths per year per 100,000 [31]. Death among our patients might have been due to a non-specific domino effect secondary to the increased health burden of the pandemic, which delayed and reduced resources in many medical domains, including cancer and other specialties. The effect is evident in Table 3, which shows that the number of patient healthcare contacts was dramatically reduced in 2020. The present study observations are well in-line with studies that reported on the higher risk of death among cancer patients due to COVID-19 [32] and the impact on breast cancer mortality [33,34].

Could the increased toxicity and death in 2020 be attributed to lung-heart alteration or reduced immunity from previous radiotherapy? In an unplanned post-hoc comparison of the five patients who died in 2020–2021 as compared with the 103 who survived through 2020, only their ages differed significantly, *p* = 0.004 (Table 5). The radiotherapy characteristics did not appear to relate with the risk of death in 2020, although nodal irradiation might have been a potential factor, *p* = 0.113.

As an alternative explanation to that of a non-specific domino effect, an overlooked COVID-19 infection may have contributed to patient death. The peak deteriorations in lung function (Figure 3) are highly suggestive of COVID-19-linked pathology, with lingering effects that could have affected mortality [35]. The reduction of daily physical activities such as walking and biking during lockdown could also have contributed to the deterioration of lung function [36]. The patients did not undergo cardiometabolic lifestyle intervention to mitigate the inactivity [37]. The drop in LVEF seen in 2020 also suggests the possibility of undiagnosed COVID-19 myocarditis [38]. With hindsight, we can see that repeated SARS-CoV-2 testing should have been considered. However, at these early times and in periods of repeated lockdowns, there were no rationale for undertaking a viral workup in patients who did not present with signs such as fever or pneumonia.

Table 3 and Figure 2 labeled the outcomes as “toxicity” for historical reasons; namely, because the design of the original trial assumed outcomes would represent toxicities. However, the use of toxicity scores should not imply that the effects must be considered as secondary to therapy. At the time of randomization and throughout the 365 days prior to randomization, 18% of patients had pulmonary “toxicity” symptoms before receiving any radiation therapy (Table 3). This is in keeping with the patient-reported outcomes (PRO) at baseline, which were well below the nominal 100% symptom-free level, most notably with a mean fatigue-free score of 64.8–69.6% and a dyspnea-free score of 84.8–88.5% among the trial’s participants [14].

Compounding these ahead-of-therapy toxicity scores, age is a major confounder in studies with long follow-up times such as the present one. The pulmonary function changes per year (Figure 3) are of the same order as those from observational studies; for example, forced vital capacity declined by 63 mL/year in the present study as compared with 56.2 mL/year (38.8 to 73.6) in a general population aged 60–102 years [39] and 65.6 mL/year in a cohort of a median age of 73.0 years [40]. The residual volume in the present study increased by 52 mL/year, which is more than the expected 5–10% increase per decade [41], whereas the CO diffusing capacity declined by 0.29/year, which is less than the 0.35–0.49/year decline in women aged ≥ 50 years who had never been smokers, as reported elsewhere [42]. Likewise, the small decline in LVEF also appears to be -related to age, in keeping with longitudinal studies of patients with preserved ejection fraction [43,44].

Concepts of mortality and survival can be confusing. In this study we were interested in when the patients died. Statistical inference would require a much larger number of patients than available here. In a survival study we would be interested in how long the patients survived. Overall survival (87.8% at 10 years, restricted mean survival 13.0 years over 14 years horizon) and breast cancer specific survival (92.6% at 10 years, restricted mean specific-survival 13.4 years over 14 years horizon) show that the current study’s population have survival rates comparable to other studies of stage I-II breast cancer (Figure 4). Other than this, the survival graphs could not identify a calendar period effect. However, TomoBreast is an ongoing study. We plan to do further survival analyses with longer follow-up periods.

Aside from the small number of patients, limitations of this study include its descriptive design. Viral serology was not assessed. The evaluation of patients was preponderantly trial related during the first five years, but thereafter the frequency of contacts and examinations were only loosely related to the trial. These later patient visits depended on the clinical circumstances, on patient symptoms, and on increasing co-morbidities with advancing age. The observed excess lung-heart functional deteriorations might have resulted from a poor-health effect of patients requiring more attention, potentially causing a less-fit bias (though this does not diminish the importance of the need for medical contact).

The strengths of the study include a build on a hybrid study design, combining a prospective randomized controlled trial on a well-defined population with the comprehensive information generated by an observational perspective. The study also builds on the eHealth data exchange platform, a Belgian public federal institution [20,21]. The platform established in the country an electronic global medical record that shared information from medical visits, including nursing observations, lab exams, imaging, general practitioner’s consults, hospitalization reports, and, most recently, the dates of receipt of the COVID-19 vaccine [22]. The platform is not specific or limited to breast cancer but enabled us to fill the gaps in the follow-up of our patients. It provided an almost real-time assessment of patients’ vital status without missing dates. The long follow-up time of no less than 10 years resulted in a considerable data yield. The use of calendar year as the backbone of the time analysis highlights a further contribution of the study: in early and intermediate-risk breast cancer, mortality occurrence over time becomes apparent only beyond 5 years, which could be an important minimum delay before survival results of early/intermediate risk breast cancer can be interpreted.

## 5. Conclusions

This long follow-up study of a cohort of breast cancer patients found a gradual decline in lung and heart function over a decade, which worsened significantly in 2020. Five non-cancer deaths, without evidence of overt SARS-CoV-2 infection, were observed around that first pandemic year. With 15 deaths observed during the entire decade preceding the pandemic, these 5 pandemic-year deaths represented one third of all the deaths that occurred during the prior ten years. This raises the concern that the deaths in the pandemic-year were premature and might have been linked to a reduction of general medical care at the time. The management of future COVID-19 epidemic waves might need to consider how to avoid a radical reduction in medical contacts with patients. The possibility that a history of nodal irradiation could have affected the outcomes deserves attention.

## Figures and Tables

**Figure 1 cancers-14-06241-f001:**
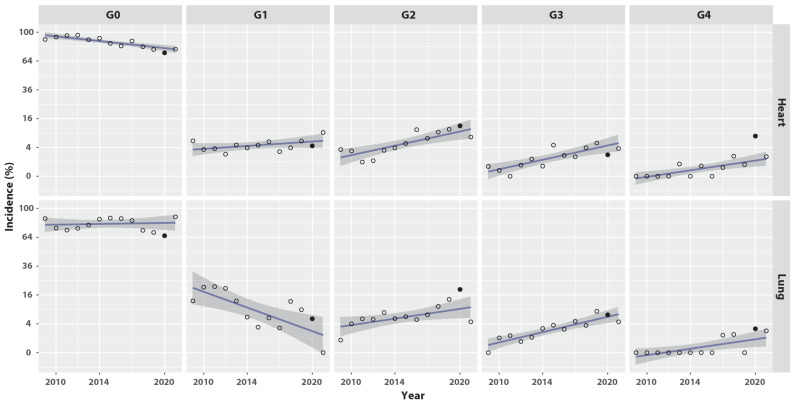
Rates of heart and lung toxicity grades over the calendar years. G0, no toxicity. G1–G4, toxicity Grade 1–4. Open circle: rate averaged on all patients observed in the year. Filled circle: rate observed in 2020, year of the COVID pandemic. Line: ordinary least squares fitted on years 2009–2021 excluding 2020. Grey band: 95% confidence interval of the least squares fit.

**Figure 2 cancers-14-06241-f002:**
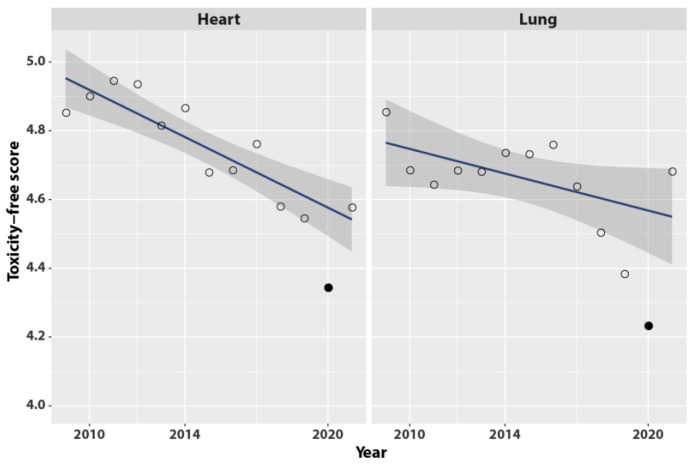
The effect by year on heart and lung toxicity-free scores. Continuous toxicity-free score scaled from 1 (worst, no freedom from toxicity) to 5 (best, fully free of toxicity), where score = (5—Grade). Open circle: score averaged on all patients observed in the year. Filled circle: score observed in 2020, year of the COVID pandemic. Line: ordinary least squares fitted on years 2009–2021 excluding 2020. Grey band: 95% confidence interval of the least squares fit.

**Figure 3 cancers-14-06241-f003:**
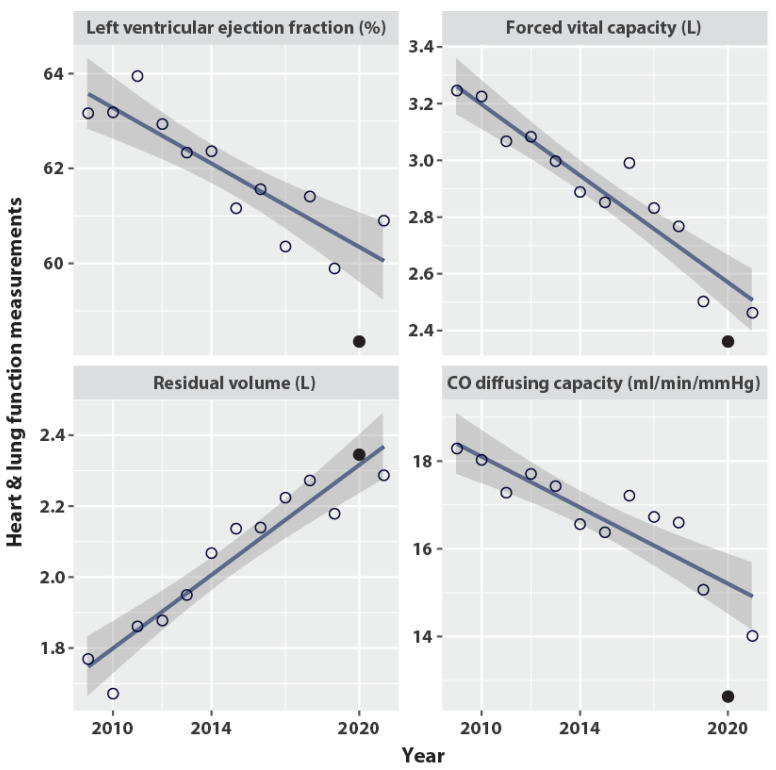
The effect by year on heart and lung function. Open circle: value of the function averaged on all patients observed in the year. Filled circle: value observed in 2020, year of the COVID pandemic. Line: ordinary least squares fitted on years 2009–2021 excluding 2020. Grey band: 95% confidence interval of the least squares fit. The slopes per year are: Left ventricular ejection fraction (%) −0.29; Forced vital capacity (L) −0.063 (=63 mL); Residual volume (L) +0.052 (=52 mL); carbon-monoxide (CO) diffusing capacity (ml/min/mmHg) −0.29.

**Figure 4 cancers-14-06241-f004:**
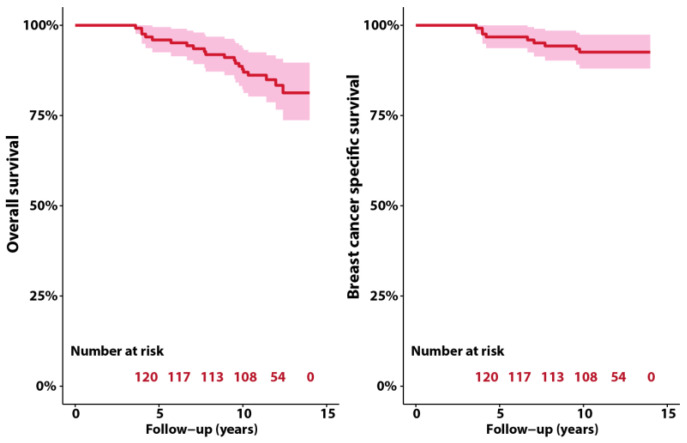
Overall survival and breast cancer specific survival of the study population.

**Table 1 cancers-14-06241-t001:** Patient characteristics, *n* = 123.

Characteristic	*n* (%)		*n* (%)
Age (years)		Mastectomy	
<50	36 (29.3%)		
≥50	87 (70.7%)	No	78 (63.4%)
Screen-detected		Yes	45 (36.6%)
No	55 (45.8%)		
Yes	65 (54.2%)	Axillary dissection	
Smoking history		No	75 (61.0%)
No	84 (68.3%)	Yes	48 (39.0%)
Yes	39 (31.7%)		
Laterality		Nodal irradiation	
Right	55 (44.7%)	No	87 (70.7%)
Left	67 (54.5%)	Yes	36 (29.3%)
Bilateral	1 (0.8%)		
Stage		Chemotherapy	
I	53 (43.1%)	No	67 (54.5%)
IIA	57 (46.3%)	Before radiation	14 (11.4%)
IIB	13 (10.6%)	Concomitant	42 (34.1%)
Progesterone receptor		Hormone therapy	
Negative	31 (25.2%)	No	17 (13.8%)
Positive	92 (74.8%)	Yes	106 (86.2%)
Neu score		Trastuzumab	
0–2	100 (82.6%)	No	110 (89.4%)
3	21 (17.4%)	Yes	13 (10.6%)

**Table 2 cancers-14-06241-t002:** Causes of death. NC, no known cancer at time of death. Case ID: reference case number linked to the shared data at https://zenodo.org/deposit/5919956 (accessed on 30 January 2022).

Year	*n* Alive in Year	*n* Died in Year	Case ID	Primary Cause of Death
2007	12	0		
2008	47	0		
2009	78	0		
2010	115	0		
2011	123	1	10	Metastatic disease
2012	122	1	28	Metastatic disease
2013	121	2	47	Metastatic disease + heart failure
			63	Metastatic disease
2014	119	1	85	Renal abscess; cognitive dysfunction; NC
2015	118	3	4	Metastatic disease
			31	Metastatic disease
			61	Metastatic disease (lung cancer)
2016	115	0		
2017	115	2	79	Unspecified; NC
			113	Metastatic disease
2018	113	1	73	Digestive pathology
2019	112	4	65	Metastatic disease
			68	Digestive pathology; NC
			76	Metastatic disease (ovarian cancer)
			84	Metastatic disease
2020	108	4	16	Digestive pathology; NC
			42	Liver failure; NC
			45	Pulmonary hypertension; bowel obstruction; NC
			103	Unspecified (toxic thyroid adenoma?); NC
2021	104	1	116	Pulmonary hypertension (right heart failure); NC

**Table 3 cancers-14-06241-t003:** Heart and lung toxicities. G0, no toxicity. G1–G4, toxicity Grade 1–4. * Half-year data.

Year	*n* Records	Heart						Lung					
		*n* Data	G0 %	G1 %	G2 %	G3 %	G4 %	*n* Data	G0 %	G1 %	G2 %	G3 %	G4 %
2007 *	70	32	100.0	0.0	0.0	0.0	0.0	28	82.1	14.3	3.6	0.0	0.0
2008	493	381	97.4	2.1	0.5	0.0	0.0	360	92.2	5.0	1.7	1.1	0.0
2009	518	433	90.1	6.0	3.5	0.5	0.0	404	86.4	12.9	0.7	0.0	0.0
2010	687	612	93.3	3.4	3.1	0.2	0.0	579	74.4	20.6	4.0	1.0	0.0
2011	531	517	95.4	3.7	1.0	0.0	0.0	512	72.3	20.9	5.5	1.4	0.0
2012	340	340	95.9	2.4	1.2	0.6	0.0	338	74.3	19.8	5.3	0.6	0.0
2013	279	277	89.9	4.7	3.2	1.4	0.7	272	78.3	12.9	7.7	1.1	0.0
2014	217	207	91.8	3.9	3.9	0.5	0.0	215	85.6	6.0	5.6	2.8	0.0
2015	195	193	85.0	4.7	5.2	4.7	0.5	194	87.1	3.1	6.2	3.6	0.0
2016	194	192	81.8	5.7	10.4	2.1	0.0	192	86.5	5.7	5.2	2.6	0.0
2017	284	275	88.0	2.9	6.9	1.8	0.4	276	84.1	2.9	6.9	4.7	1.4
2018	317	256	80.9	3.9	9.4	3.9	2.0	254	72.0	12.6	10.2	3.5	1.6
2019	220	151	77.5	6.0	10.6	5.3	0.7	147	69.4	8.8	13.6	8.2	0.0
2020	134	90	73.3	4.4	12.2	2.2	7.8	73	65.8	5.5	19.2	6.8	2.7
2021 *	168	54	77.8	9.3	7.4	3.7	1.9	44	88.6	0.0	4.5	4.5	2.3
All years	4647	4010	90.2	4.0	4.1	1.2	0.4	3888	79.5	12.6	5.5	2.1	0.3

**Table 4 cancers-14-06241-t004:** Heart and Lung function. * Half-year data. SD, standard deviation. LVEF, left ventricular ejection fraction; FVC, forced vital capacity; RV, residual volume; DLCO, carbon-monoxide diffusing capacity.

Year	*n* Records	LVEF (%)			FVC (L)			RV (L)			DLCO (ml/min/mmHg)		
		*n* Data	Mean	SD	*n* Data	Mean	SD	*n* Data	Mean	SD	*n* Data	Mean	SD
2007 *	70	62	62.3	4.7	46	3.9	0.5	46	1.7	0.4	46	19.8	1.9
2008	493	446	63.8	4.4	393	3.6	0.7	391	1.9	0.5	391	18.5	3.3
2009	518	467	63.2	5.1	449	3.2	0.7	438	1.8	0.6	446	18.3	3.5
2010	687	587	63.2	4.5	621	3.2	0.6	612	1.7	0.5	612	18.0	3.2
2011	531	441	63.9	4.9	474	3.1	0.6	470	1.9	0.5	469	17.3	3.3
2012	340	216	62.9	4.4	260	3.1	0.6	256	1.9	0.5	256	17.7	3.4
2013	279	152	62.3	4.8	184	3.0	0.6	179	1.9	0.5	179	17.4	3.5
2014	217	121	62.4	4.6	117	2.9	0.7	111	2.1	0.6	113	16.6	3.6
2015	195	116	61.2	4.7	97	2.9	0.7	87	2.1	0.5	92	16.4	3.6
2016	194	104	61.6	5.7	104	3.0	0.6	99	2.1	0.5	101	17.2	3.8
2017	284	103	60.4	5.2	148	2.8	0.8	135	2.2	0.5	140	16.7	4.4
2018	317	104	61.4	6.8	140	2.8	0.8	123	2.3	0.4	127	16.6	4.0
2019	220	78	59.9	6.9	81	2.5	0.8	65	2.2	0.5	68	15.1	4.8
2020	134	44	58.4	8.2	46	2.4	0.7	36	2.3	0.6	43	12.6	5.0
2021 *	168	10	60.9	7.6	13	2.5	0.7	10	2.3	0.5	10	14.0	5.1
All years	4647	3051	62.8	5.1	3173	3.1	0.7	3058	1.9	0.5	3093	17.6	3.6

**Table 5 cancers-14-06241-t005:** Baseline characteristics of patients who survived or died in 2020–2021.

Baseline Characteristic	Alive through 2020 *n* = 103	Died in 2020–2021 *n* = 5	*p*-Value
Age (years)			0.004
Mean (SD)	55.3 (10.9)	70.0 (7.6)	
Karnofsky Performance Status			0.816
*n* missing	5	0	
Mean (SD)	94.8 (7.4)	94.0 (8.9)	
Weight (kg)			0.424
Mean (SD)	67.8 (12.1)	63.4 (10.3)	
Smoking history			0.600
No	71 (68.9%)	4 (80.0%)	
Yes	32 (31.1%)	1 (20.0%)	
Laterality			0.474
Right	45 (43.7%)	3 (60.0%)	
Left (1 case bilateral)	58 (56.3%)	2 (40.0%)	
Stage			0.805
I	56 (54.4%)	3 (60.0%)	
II	47 (45.6%)	2 (40.0%)	
Estrogen receptor positive			0.686
No	14 (13.6%)	1 (20.0%)	
Yes	89 (86.4%)	4 (80.0%)	
Progesterone receptor positive			0.864
No	24 (23.3%)	1 (20.0%)	
Yes	79 (76.7%)	4 (80.0%)	
Mastectomy			0.442
No	65 (63.1%)	4 (80.0%)	
Yes	38 (36.9%)	1 (20.0%)	
Axillary Dissection			0.298
No	65 (63.1%)	2 (40.0%)	
Yes	38 (36.9%)	3 (60.0%)	
Randomization arm			0.709
Conventional radiotherapy	53 (51.5%)	3 (60.0%)	
Tomotherapy	50 (48.5%)	2 (40.0%)	
Radiotherapy (RT) boost			0.467
No (= RT chest wall only)	37 (35.9%)	1 (20.0%)	
Yes (= RT breast conserving surgery)	66 (64.1%)	4 (80.0%)	
Nodal irradiation			0.113
No	75 (72.8%)	2 (40.0%)	
Yes	28 (27.2%)	3 (60.0%)	
Chemotherapy			0.277
No	57 (55.3%)	4 (80.0%)	
Yes	46 (44.7%)	1 (20.0%)	
Hormone therapy			0.397
No	13 (12.6%)	0 (0.0%)	
Yes	90 (87.4%)	5 (100.0%)	
Trastuzumab therapy			0.441
No	92 (89.3%)	5 (100.0%)	
Yes	11 (10.7%)	0 (0.0%)	

## Data Availability

The data presented in this study are available on Zenodo: https://zenodo.org/deposit/5919956 (accessed on 30 January 2022) with reserved doi:10.5281/zenodo.5919956.

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
