# Peer review of "Lung-Heart Outcomes and Mortality through the 2020 COVID-19 Pandemic in a Prospective Cohort of Breast Cancer Radiotherapy Patients"

_cancers, 2022, doi:10.3390/cancers14246241_

Round 1

Reviewer 1 Report

Line 54-55

You do not need this “Data management resumed in mid-2021, when the principal investigator travelled back to Belgium to query on patient’s status. The question immediately arose on how much the pandemic had affected the patterns”

In line 61 you clearly specify the purpose of the study. “In order to investigate the possibility of indirect effects of COVID-19 in……”

What I miss in the discussion is a possible link between those who received radiation therapy in chest area and an increased Covid-19 linked pathology. In your study there is a very strong increase in all deaths following the pandemic. You nicely refer to the Belgian Cancer registry where they found an 33% increase in mortality in the first half of 2020. In your study the increased mortality following the pandemic is 200-400% (pre pandemic 1-2 deaths a year, post pandemic 4 deaths a year). 

In conclusion you could mention that the mortality is higher than expected compared to other cancer patients. Patients who have received radiation therapy towards chest area might need closer follow up in future covid 19 epidemic waves. 

Author Response

Point 1: Line 54-55

You do not need this “Data management resumed in mid-2021, when the principal investigator travelled back to Belgium to query on patient’s status. The question immediately arose on how much the pandemic had affected the patterns”

In line 61 you clearly specify the purpose of the study. “In order to investigate the possibility of indirect effects of COVID-19 in……”

 Response 1: The revision removes the Line 54-55 sentences.

Point 2: What I miss in the discussion is a possible link between those who received radiation therapy in chest area and an increased Covid-19 linked pathology.

Response 2: The revised discussion adds Table 5 and inserts the following paragraph:

Could the increased toxicity and death in 2020 be attributed to lung-heart alteration or reduced immunity from previous radiotherapy? In an unplanned post-hoc comparison of the 5 patients who died in 2020–2021 as compared with the 103 who survived through 2020, only age differed significantly, P=0.004 (Table 5). The radiotherapy characteristics did not appear to relate with the risk of death in 2020, although nodal irradiation might have been a potential factor, P=0.113.

Point 3: In your study there is a very strong increase in all deaths following the pandemic. You nicely refer to the Belgian Cancer registry where they found an 33% increase in mortality in the first half of 2020. In your study the increased mortality following the pandemic is 200-400% (pre pandemic 1-2 deaths a year, post pandemic 4 deaths a year).

In conclusion you could mention that the mortality is higher than expected compared to other cancer patients. Patients who have received radiation therapy towards chest area might need closer follow up in future covid 19 epidemic waves.

Response 3: As per Point 2, the revised discussion added Table 5 and discussed there were no significant factor other than age. However, nodal irradiation might have been a potential factor.

The conclusion adds the following sentence: The possibility that a history of nodal irradiation could have affected the outcomes deserves attention.

Reviewer 2 Report

Dear authors
Thank you for the conducted valuable research.
Here are a few comments which needs to be clarified within the paper:
- One of the most important outcomes, which has been evaluated by authors, is "survival" which is not mentioned in the title of manuscript and needs to be reflected.
- Some of the explained results are not related directly to the current study's objectives. For example, this study aimed on the outcomes within 2020 (based on the title) and discussing the details before this time point would be confusing.
- Introduction is not written as routine, and actually it's mainly explain the ongoing prospective study, which could be moved to the methods section. Therefore, I would suggest that the authors mention the importance of the topic and available knowledge gap within the introduction section.
- The coulum case ID could be removed from Table 2.
- The quality of Figures need to be improved.
- It could be an idea that the authors separate the survival analysis to overall survival and disease specific survival.
- The authors need to add a statistical plan (according to the objectives) in the methods section and present the results in the same order.
- Discussion needs to be updated; as the reached results were not compared with other studies regarding the mortality of breast cancer patients after the COVID-19 pandemic. These 2 references could be example to be mentioned and discussed within the discussion section:
https://pubmed.ncbi.nlm.nih.gov/34258611/
https://pubmed.ncbi.nlm.nih.gov/33399255/
Or Heart metabolic:
https://pubmed.ncbi.nlm.nih.gov/34204528/
- Methods section could be divided to different sub-sections in order to be easier to follow for the readers.
- Perhaps a separate paragraph in methods section explaining the Belgium healthcare policy on management of breast cancer patients would help.

Author Response

Point 1: One of the most important outcomes, which has been evaluated by authors, is "survival" which is not mentioned in the title of manuscript and needs to be reflected.

Response 1: We thank reviewer who pointed out the inconsistency. The revised title is:

Lung-heart outcomes and mortality through the 2020 COVID-19 pandemic in a prospective cohort of breast cancer radiotherapy patients.

The revised Abstract replaces “survival” with “mortality”.

Point 2: Some of the explained results are not related directly to the current study's objectives. For example, this study aimed on the outcomes within 2020 (based on the title) and discussing the details before this time point would be confusing.

Response 2: The revised manuscript inserts the explanation why we had to look at previous years.

Revised Methods, section 2.8. Longitudinal analysis of continuous measurements:

For the lung-heart outcomes, we already knew that the function steadily deteriorated over time [23]. Consequently, in order to detect whether an additional change occurred in the particular year 2020, we could not compare with a fixed time-point in the past but had to take into account what would have normally been expected if the pandemic did not occur. We applied ordinary least squares regression of toxicity grades (converted to fAE) and heart-pulmonary measurements over the years, excluding 2020. Thereafter we tested whether the actual measurements and toxicities observed in 2020 significantly departed from the regression as outliers.

Point 3: Introduction is not written as routine, and actually it's mainly explain the ongoing prospective study, which could be moved to the methods section. Therefore, I would suggest that the authors mention the importance of the topic and available knowledge gap within the introduction section.

 Response 3: We thank reviewer for help to improve by stating the importance and filling a knowledge gap. The revised Introduction has been structured as follows:

The pandemic was devastating.

Cancer patients are vulnerable. But longitudinal quantitative studies are scarce or limited.

TomoBreast was designed with lung and heart function as core concern.

It provides the most comprehensive database of longitudinal lung and heart observations.

The data is important, we make it openly available.

Point 4: The coulum case ID could be removed from Table 2.

Response 4: We believe that the column “Case ID” is necessary to researchers who would want to explore our data. The revised caption of Table 2 inserts:

“Case ID: reference case number linked to the shared data at https://zenodo.org/deposit/5919956.”

Point 5: The quality of Figures need to be improved.

 Response 5: All figures have been redrawn, and also correct unnoticeable paste error in the original manuscript.

Point 6: It could be an idea that the authors separate the survival analysis to overall survival and disease specific survival.

Response 6: The revised manuscript distinguishes the mortality table, versus survival analysis.

The revised Methods inserts sub-section 2.9 Mortality table and survival:

The time variable required for survival analysis is the interval between an observation date and a given time origin, i.e. the follow-up year is embedded in the survival model. Unless all patients in a cohort are accrued on the same date, in which case the follow-up period equates the time elapsed, conventional survival analysis cannot detect the effect of a particular follow-up year. The present study simply tabulated the deaths observed over the calendar years, without significance testing. Mortality in the study refers to the crude number of deaths.

For discussion purpose –not as study objectives– the overall survival from date of randomization to death of any cause, and the breast cancer specific survival, from date of randomization to death from breast cancer, were computed using the Kaplan-Meier product-limit method [26].

The revised Discussion inserts the paragraph:

Concepts of mortality and survival can be confusing. In this study we were interested about when did the patients die. Statistical inference would require a much larger number of patients than available here. In a survival study we would be interested in how long the patients survived. Overall survival (87.8% at 10 years, restricted mean survival 13.0 years over 14 years horizon) and breast cancer specific survival (92.6% at 10 years, restricted mean specific-survival 13.4 years over 14 years horizon) show that the current study population have survival comparable to other studies of stage I-II breast cancer (Figure 4). Other than that, the survival graphs could not identify a calendar period effect.

Point 7: The authors need to add a statistical plan (according to the objectives) in the methods section and present the results in the same order.

 Response 7: We are thankful for the suggestion to improve clarity. Methods have been restructured with sub-sections, specifically for statistics:

2.8. Longitudinal analysis of continuous measurements

2.9. Mortality table and survival

2.10. Statistical implementation

Point 8: Discussion needs to be updated; as the reached results were not compared with other studies regarding the mortality of breast cancer patients after the COVID-19 pandemic. These 2 references could be example to be mentioned and discussed within the discussion section:

https://pubmed.ncbi.nlm.nih.gov/34258611/

https://pubmed.ncbi.nlm.nih.gov/33399255/

Or Heart metabolic:

https://pubmed.ncbi.nlm.nih.gov/34204528/

Response 8: We thank reviewer for the references. The revised discussion inserts the relevant citations:

The present study observations are well in-line with studies that reported on the COVID-19 higher risk of death among cancer patients [30] and the impact on breast cancer mortality [31,32].

The patients did not undergo cardiometabolic lifestyle intervention to mitigate the inactivity [35].

Point 9: Methods section could be divided to different sub-sections in order to be easier to follow for the readers.

 Response 9: We thank reviewer suggestion to improve the readability. Methods have been restructured into sub-sections:

2.1. Description of the cohort

2.2. Clinical data collection

2.3. Toxicity grades

2.4. Freedom from adverse event (fAE) scores

2.5. Echocardiography assessment

2.6. Lung function assessment

2.7. Interpolation and handling of missing data

2.8. Longitudinal analysis of continuous measurements

2.9. Mortality table and survival

2.10. Statistical implementation

Point 10: Perhaps a separate paragraph in methods section explaining the Belgium healthcare policy on management of breast cancer patients would help.

Response 10: We thank reviewer for pointing to a source of confusion. The manuscript has been revised:

In Methods, section 2.2 Clinical data collection: Patients' information was abstracted from individual patient's electronic medical files and from the Belgian federal eHealth platform [20-22],

where references [20-22] link to the eHealth information.

In Discussion: The revision replaces “system” with the dedicated term “platform”, and rewrites the sentence as follows: “The study also builds on the eHealth data exchange platform, a Belgian public federal institution [20,21]. The platform established in the country an electronic global medical record…”

In Discussion last paragraph: “The platform is not specific or limited to breast cancer”.

Round 2

Reviewer 2 Report

Dear Authors
Thanks for the valuable revision throughout the whole manuscript. It is now more structured and much easier to follow the section within the whole text.
Just a few minor points:
- Use past tense in study objectives (the present study aimed to...)
- The sub-section named "Mortality table and survival" could just change to "Survival analysis" as it also includes the information regarding mortality.
- One of overall or disease-specific survival results could be added to abstract.
- Since it is an ongoing study, it could be relevant to add a prospective in discussion section regarding the further analysis with longer follow-up period (for survival analysis).

Author Response

We are most grateful to reviewer’s comments to further improve the manuscript.

Point 1:

- Use past tense in study objectives (the present study aimed to...)

Response 1:

The past tense has been applied as suggested: “The present study aimed to “...

Point 2:

- The sub-section named "Mortality table and survival" could just change to "Survival analysis" as it also includes the information regarding mortality.

Response 2:

The sub-section has been renamed to "Survival analysis" as suggested.

Point 3:

- One of overall or disease-specific survival results could be added to abstract.

Response 3:

The Abstract adds the overall survival information: “At 12.0 years median follow-up, 103 patients remained alive; 10-years overall survival was 87.8%”. The next and last abstract’ sentence have been shortened to keep word count at 200: “In 2007-2019, 15 patients died…”, and “Deaths in that year totaled one-third…”.

 Point 4:

- Since it is an ongoing study, it could be relevant to add a prospective in discussion section regarding the further analysis with longer follow-up period (for survival analysis).

Response 4:

The Discussion adds (Lines 329-331): “However, TomoBreast is an ongoing study. We plan to do further survival analyses with longer follow-up”.
